



# Obtaining reliable localizations with Time Reverse Imaging: limits to array design, velocity models and signal-to-noise ratios

Claudia Werner[1,2] and Erik H. Saenger[1,2]

[1]International Geothermal Center Bochum, Lennershofstr. 140, 44801 Bochum, Germany
[2]Ruhr-University Bochum, Universitätsstr. 150, 44801 Bochum, Germany

**Correspondence:** Claudia Werner (claudia.werner@hs-bochum.de)

**Abstract.** Time Reverse Imaging (TRI) is evolving into a standard technique for localizing and characterizing seismic events. In recent years, TRI has been applied to a wide range of applications from the lab scale over the field scale up to the global scale. No identification of events and their onset times is necessary when localizing events with TRI. Therefore, it is especially suited for localizing quasi-simultaneous events and events with a low signal-to-noise ratio. However, in contrast to more regularly

applied localization methods, the prerequisites for applying TRI are not sufficiently known.

To investigate the significance of station distributions, complex velocity models and signal-to-noise ratios for the localization quality, numerous simulations were performed using a finite difference code to propagate elastic waves through three-dimensional models. Synthetic seismograms were reversed in time and re-inserted into the model. The time-reversed wavefield backpropagates through the model and, in theory, focuses at the source location. This focusing was visualized using imaging

conditions. Additionally, artificial focusing spots were removed with an illumination map specific to the setup. Successful localizations were sorted into four categories depending on their reliability. Consequently, individual simulation setups could be evaluated by their ability to produce reliable localizations.

Optimal inter-station distances, minimum apertures, relations between array and source location, heterogeneities of inter-station distances and total number of stations were investigated for different source depth as well as source types. Additionally,

the quality of the localization was analysed when using a complex velocity model or a low signal-to-noise ratio.

Finally, an array in Southern California was investigated for its ability to localize seismic events in specific target depths while using the actual velocity model for that region. In addition, the success rate with recorded data was estimated.

Knowledge about the prerequisites for using TRI enables the estimation of success rates for a given problem. Furthermore, it reduces the time needed for adjusting stations to achieve more reliable localizations and provides a foundation for designing

arrays for applying TRI.

## 1 Introduction

The localization and characterization of seismic events in the subsurface is crucial for understanding physical processes in the Earth. Well-established methods are able to localize most seismic events in a fast and reliable manner; but they rely on identifiable onsets of events. Time Reverse Imaging (TRI) is a method especially suited for localizing and characterizing events





which are indistinguishable in traces because they occur quasi-simultaneously or are superposed by noise. The prerequisites for more regularly applied localization methods are very well known. However, which station distributions, which degree of complexity in the velocity model and which level of noise hinder or enhance localizations with TRI is not sufficiently known. Therefore, this study systematically tests different station distributions for their localization capabilities while considering

complex velocity models and low signal-to-noise ratios.

TRI uses the whole recorded waveform rendering the identification of events and their onsets obsolete. It can be applied as long as the wave propagation can be described with a time-invariant wave equation. Seismic traces are reversed in time and backpropagated through a medium until they focus on the initial event location. Imaging Conditions are used to visualize aspects of the backpropagating wavefield and obtain the localization point.

TRI has been applied in earth sciences as well as medical sciences for some time (Fink et al., 1999). In recent years multiple studies have shown that TRI is an easy-to-use and reliable localization tool: It has been used to retrieve source information on various scales from the lab scale in non-destructive testing (Saenger, 2011; Anderson et al., 2011; Harker and Anderson, 2013; Kocur et al., 2016) over the field scale, for example in volcanic tremor (Lokmer et al., 2009) and non-volcanic tremor applications (Larmat et al., 2009; Horstmann et al., 2015) and above hydrocarbon reservoirs (Steiner et al., 2008), up to the

global scale (Larmat et al., 2006, 2008). However, to apply TRI, a fairly accurate velocity model is needed to accurately backpropagate the wavefield. With increasing availability of high resolution large three-dimensional velocity models and the knowledge about prerequisites, TRI has the potential to localize seismic events, which could not be localized reliably thus far, in a wide range of applications.

### 1.1 Restrictions to the localization capabilities of TRI

The estimation of localization qualities is one major challenge when applying TRI. A common approach is, therefore, to perform a preliminary synthetic study to test if the given velocity model and station distribution enable reliable localizations. If the synthetic study fails, the setup is adjusted until either the study is abandoned or a sufficiently reliable result is achieved. This adjustment phase can be time-consuming because there are multiple characteristics appearing at once that may hinder localizations.

Numerous characteristics of where stations are placed at the surface seem to influence the chance of success with TRI significantly. In theory, TRI works with only one station (Montagner et al., 2012). Nevertheless, in most cases more stations are needed to obtain a stable result. Therefore, the total number of stations is often considered to have a major influence on the success or failure of localizations (Kremers et al., 2011; Horstmann et al., 2015). However, numerous reports state that a smaller amount of stations works just as well (Gajewski and Tessmer, 2005; Steiner et al., 2008; Lokmer et al., 2009; Larmat

et al., 2009).

The aperture of the array seems to be another characteristic of the station distribution influencing localization quality. Steiner et al. (2008) and Lokmer et al. (2009) report false localizations in the two-dimensional case when the aperture of the array is limited. Artman et al. (2010) emphasize the importance of a sufficiently large aperture to get a spatially focused localization.





Distances between the individual stations and the homogeneity of these inter-station distances seem to be important as well (Lokmer et al., 2009). Furthermore, the azimuthal gap, which is the angle between two stations viewed from the epicentre location, is introduced by Horstmann et al. (2015). It is an indirect measurement for the heterogeneity of inter-station distances. Horstmann et al. (2015) observe an enhanced localization result if the maximum azimuthal gap is small. Lokmer et al. (2009)

and Horstmann et al. (2015) also note a different quality of focusing for sources in different depths while using the same stations.

In addition to the placement of stations, the velocity model can influence the failure or success of TRI. An erroneous velocity model may inhibit the localization of real events while a rather complex velocity model may hinder the localization already during preliminary synthetic tests. As long as errors are small enough, TRI appears to be successful with a false velocity model

(Gajewski and Tessmer, 2005; Lokmer et al., 2009; Saenger, 2011). Similarly, a smoothed velocity model does not significantly influence the results (Gajewski and Tessmer, 2005). Nevertheless, necessary simplifications may inhibit localizations of events with TRI (Artman et al., 2010; Horstmann et al., 2015). Common simplifications include a constant ratio of p-wave velocity to s-wave velocity or a constant density. Additionally, complex velocity structures may alter the final image produced with TRI (Larmat et al., 2009).

A high level of noise in the traces may inhibit localizations with TRI, even when the array and the velocity model are sufficiently well. In theory, TRI works even with a very low signal-to-noise ratio because noise is random and will not superpose constructively to form a focus during the backpropagation of the wavefield. Gajewski and Tessmer (2005) show traces with events not distinguishable from the noise and TRI is still able to reconstruct the source location. Witten and Artman (2011) create synthetic data with signal-to-noise ratios as low as 0.25 and show successful localizations. Nevertheless, they observe a

decrease in localization accuracy with lower signal-to-noise ratios.

## 1.2 Objective of this study

This study aims at finding station distributions producing reliable localizations with TRI. This is crucial to estimate the success rate of TRI with a given set of stations. Additionally, the time needed to adjust station distributions may be decreased. Furthermore, the prerequisites of the method should be known when designing an array for using TRI. Therefore, the influence of the

station distribution, the complexity of the velocity model and the rate of noise on the localization quality were investigated.

We performed numerous simulations to systematically analyse different station distributions and their influence on the localization quality of sources in different depths as well as of different source types. The focus was thereby on the distance between receivers, the symmetry of the array in relation to the source position, the azimuthal gap between receivers and the number of stations. Simulations were performed first with a homogeneous velocity model and afterwards with a complex

velocity model. Localization quality was investigated while the velocity model is known and when it is not known correctly. We investigated the ability of TRI to cope with very low signal-to-noise ratios. To complete the study, we applied the guidelines found with the methodical tests to an actual example in Southern California. The ability to localize events in a target depth was investigated while using the existing array as well as the actual velocity model for that region by Zeng et al. (2016).



With the example in Southern California we demonstrate how to assess the success rate of localizations with TRI using a synthetic study. Additionally, the station distribution was altered to enhance localization quality. We give estimates in which part of the model reliable localizations can be expected. Additionally, we demonstrate how to design an array, which produces reliable localizations.

All simulations were performed using a finite difference scheme to propagate elastic waves through three-dimensional models. The advantage of a purely synthetic study is the suppression of unknown side effects. Therefore, all influences on the localization were controlled to investigate each parameter individually.

## 2 The method of Time Reverse Imaging (TRI)

Time Reverse Imaging (TRI) uses the whole waveform of recorded seismograms to localize and characterize seismic events.
The method consists of three steps following the workflow introduced by Saenger (2011) and modified by Witten and Artman (2011): Reversal of individual traces in time, backpropagation of time-reversed wavefield and elimination of artefacts impinged by velocity structure and model setup. Results are afterwards visualized using suitable imaging conditions. In the following, the adaptation of this workflow to this study is described in detail.

The method of TRI was investigated using synthetic seismograms created by forward simulations of the propagation of
elastic waves through a medium. The obtained seismograms were not altered besides flipping them in time. Montagner et al. (2012) use binarized seismograms to demonstrate that TRI is based on the coherency of the phases in the seismograms and not on the amplitudes. Therefore, the phase information in the recorded seismograms should be kept as close to the original as possible and any filter should be a zero-phase filter to prevent phase shifts in the time domain. Additionally, all traces need to be time-synchronous.

The time-reversed seismograms were reinserted into the model domain at the exact locations they were recorded at. The receivers of the forward simulation act as sources in the time-reversed simulation. In this study, receivers always mean stations at the surface which act as receivers in forward simulations and sources in reverse simulations. The time-reversed wavefield backpropagates through the model and collapses at the initial source location.

Imaging conditions highlight specific characteristics of the backpropagating wavefield. They were used to visualize the point
in space and time where the wavefield focuses. In this numerical study, the event time was known and the position of foci are compared to the source position used in the forward simulation to determine the quality of localizations.

Apart from the focus spot at the initial source location, parts of the velocity structure and the model domain itself may be highlighted by artificial focusing spots. Usually, artificial focusing spots appear around the receiver positions at the surface of the model. Additionally, special velocity structures, like low velocity zones, may cause artificial focusing (Larmat et al.,
2009). To eliminate those artefacts, the workflow introduced by Witten and Artman (2011) was used. Images produced with the specific imaging conditions were divided by images produced by the backpropagation of random noise. To keep amplitudes and frequency content the same, recorded seismograms were backpropagated through the model not time-reversed. The not time-reversed wavefield cannot focus on the initial source location and acts as a noise field. The result is an illumination map





of the model highlighting areas where focusing occurs solely due to the used velocity structure and the model setup. When dividing the results by this illumination map ideally only focusing spots created by interferences of the time-reversed traces remain in the image.

## 2.1 Simulation of Wave Propagation in Elastic Media

The wave propagation in three-dimensional elastic media for the forward as well as the backward simulation was performed using a finite difference code developed by Saenger et al. (2000). A finite-difference operator of length two was used. The model boundaries at the bottom and at the sides of the models were absorbing boundaries after Clayton and Engquist (1980) and the top boundary was a free surface that is incorporated as a vacuum layer.

A non-volcanic tremor application in Southern California was used as a real-life example. Therefore, the models used in this
study were set up to be numerically stable when using the receivers used in Horstmann et al. (2015) and the updated velocity model by Zeng et al. (2016) that covers the same region as in Horstmann et al. (2015). All models were $130\,\mathrm{km}$ by $120\,\mathrm{km}$ by $28\,\mathrm{km}$ large. The grid spacing was set to $0.1\,\mathrm{km}$ to balance accuracy and stability in the simulations, while the time step was set to $0.01\,\mathrm{s}$.

Sources in the forward simulations were implemented as moment tensor sources. An explosion source (only $M_{xx} = M_{yy} =$
$M_{zz}$ were non-zero) or a strike-slip source (only $M_{xy}$ was non-zero) was used. As a source signal, the negative normalized second derivative of the Gaussian function, also called a Ricker wavelet, was used. The Ricker wavelet was implemented, following Shearer (2009), as

$$R(t) = (1 - 2\pi^2 f_p^2 t^2) \cdot exp(-\pi^2 f_p^2 t^2), \tag{1}$$

where $t$ denotes the time and $f_p$ the peak frequency of the wavelet. For all simulations in this study the peak frequency was set
to $f_p = 1.75\,\mathrm{Hz}$ which results in a maximum frequency of about $f_{max} = 5\,\mathrm{Hz}$. In this frequency range non-volcanic tremor is usually observed (Horstmann et al., 2015). The wavelet was centred around $t = 0\,\mathrm{s}$.

The source position was set to $x = -1.8\,\mathrm{km}$ and $y = -28.3\,\mathrm{km}$, which corresponds to an existing recorded earthquake that was chosen randomly and represents one possible source location. The earthquake occurred at the 28th of June 2011 at 14:10 and had a magnitude of 1.68 (NCEDC, 2014). The three depths used for the sources are: $z_1 = 5\,\mathrm{km}, z_2 = 11.9\,\mathrm{km}$ and
$z_3 = 22\,\mathrm{km}$. The deepest source represents the depth where non-volcanic tremor occurs. The medium depth is the depth of the example earthquake while the shallow source represents a shallow end-member and is not geologically relevant in the considered region in Southern California.

## 2.2 Imaging Conditions

Imaging conditions were used to visualize specific characteristics of the backpropagated wavefield. They were calculated as
the maximum value of specific characteristics of the wavefield at each point $x$ of the model domain over the whole simulation time, from $t = 0\,\mathrm{s}$ to $t = T$. A perfect result would be a single gridpoint with the highest value of the specific imaging condition at the initial source location and very low values everywhere else.



Numerous imaging conditions have been proposed in the past. Here we summarise characteristics of the wavefield that can be used to calculate imaging conditions. The most intuitive imaging condition uses the maximum amplitude of displacement (e.g. used in Gajewski and Tessmer (2005), Larmat et al. (2009) and Saenger (2011)). The maximum particle velocity is also a suitable imaging condition (Steiner et al., 2008; Lokmer et al., 2009). Alternatively, the divergence and the curl field can
be used for imaging conditions that are sensitive to p-waves and s-waves, respectively (Larmat et al., 2009; Lokmer et al., 2009; Horstmann et al., 2015). The strain field has the potential to directly reveal source characteristics (Larmat et al., 2009; Kremers et al., 2011). Additionally, energy densities can be computed from the parameters mentioned above (Artman et al., 2010; Saenger, 2011).

In this study, we focused on the four imaging conditions proposed by Saenger (2011): The maximum particle displacement
$\mathbf{I}_d$ was calculated by

$$\mathbf{I}_d(\boldsymbol{x}) = \max_{t \in [0,T]} \|\boldsymbol{u}(\boldsymbol{x},t)\|, \tag{2}$$

with $\boldsymbol{u}(\boldsymbol{x},t)$ being the displacement at each point $\boldsymbol{x}$ in the model at each time $t$.

The maximum p- and s-wave energy density imaging conditions ($\mathbf{I}_p$ and $\mathbf{I}_s$) were calculated by separating the wavefield in a divergence field and a curl field, followed by the calculation of the energy densities after Dougherty and Stephen (1988):

$$\mathbf{I}_p(\boldsymbol{x}) = \max_{t \in [0,T]} (\lambda + 2\mu)[\nabla \cdot \boldsymbol{u}(\boldsymbol{x},t)]^2, \tag{3}$$

$$\mathbf{I}_s(\boldsymbol{x}) = \max_{t \in [0,T]} \mu[\nabla \times \boldsymbol{u}(\boldsymbol{x},t)]^2. \tag{4}$$

$\lambda$ and $\mu$ represent the Lamé parameters. The maximum total energy density imaging condition $\mathbf{I}_e$ was based on the multiplication of stresses and strains at every point in the model domain:

$$\mathbf{I}_e(\boldsymbol{x}) = \max_{t \in [0,T]} \sum_i \sum_j [\sigma_{ij}(\boldsymbol{x},t)\varepsilon_{ij}(\boldsymbol{x},t)]. \tag{5}$$

$\sigma_{ij}(\boldsymbol{x},t)$ are the stresses and $\varepsilon_{ij}(\boldsymbol{x},t)$ the strains at each point $\boldsymbol{x}$ in the model domain at each time $t$.

These four imaging conditions were calculated with separate characteristics of the wavefield. Therefore, they enable the independent inspection of the focus point.

## 2.3 Evaluation of localization success

In this study, we want to find prerequisites for the application of TRI. Therefore, it is necessary to be able to evaluate the
reliability of the focusing. TRI claims to work with no a priori information about the number of events and their position in time and space, which implies that results from this method should enable the localization of events unambiguously. The origin time was known for all simulations and we used only one source per simulation. Therefore, the focus was on the quality of the localization as a position in space.

The localization of an event can be found by looking for the point with the maximum value in each imaging condition.
Because the wavefield was inserted only at discrete points at the surface during the backpropagation, the wavefront needs to





heal before being able to focus on the source location. The depth in which the wavefront is healed depends on the discretization of the emitted wavefield (Witten and Artman, 2011). In our simulations artefacts remained after dividing the results by the illumination map. These artefacts were close to the surface and had higher amplitudes than the localization. We observed that on average at a depth of one p-wave wavelength sampling artefacts stopped interfering with the localization of events.

Therefore all values above the depth of one p-wave wavelength were set to zero. With the used frequency range and velocity models, this boundary depth was $2.3\,\mathrm{km}$ beneath the free surface.

Localizations found while excluding the shallowest parts of the model were then compared to the initial source location set in the forward simulation. A successful localization was allowed to deviate less than $1.2\,\mathrm{km}$ from the initial source localization. Localizations deviating further than the threshold from the source location were considered not successful. The deviation was

calculated as the mean of the deviation in all three directions. We did not define a source area with a certain radius around the source location but rather allowed localizations to be shifted a greater distance in one direction while being very close to the source in the other two directions. The deviation threshold of $1.2\,\mathrm{km}$ was found empirically but coincides with approximately half the p-wave wavelength. The maximum values found in the imaging conditions were either inside this range or far away from it.

After the step described above, simulations were sorted into either producing localizations close to the source location or far away from it. The advantage of TRI is, however, to not only provide point coordinates of the localization. The imaging conditions can be visualized as well. In order to compare simulations and imaging conditions, each imaging condition was normalized in itself resulting in the maximum value (the localization) having the value one. When visualizing the normalised imaging conditions, it proved to be useful to create separate plots containing all points with a certain fraction of the maximum

value. Plots were created for values in incremental steps of 0.1, corresponding to a tenth of the maximum. By viewing all points of the imaging conditions with a specific fraction, successful localizations could be assessed further. An example of this plot type is shown in Fig. 1. The top two plots show imaging conditions for the total energy density imaging condition $\mathbf{I}_e$ for two different values (0.6 and 0.9). The bottom two plots show the same for the p-wave energy density imaging condition $\mathbf{I}_p$.

Successful localizations were viewed as shown in Fig 1 and sorted into four categories. Figure 2 is a one-dimensional

representation outlining relevant aspects of the four categories. This representation is equivalent to viewing a one-dimensional profile through the localization and noting the values of the imaging conditions along this profile.

Category I included the most reliable localizations. There was one peak inside the source area corresponding to the localization of the event. Other peaks were below 0.6, which is equivalent to $60\,\%$ of the amplitude of the highest peak. When visualizing the whole model domain and looking at all points with a value of 0.6, there would be only one focus. The plots

showing $\mathbf{I}_e$ in Fig. 1 would be categorised into category I.

Category II was similar to Category I but one (or more) secondary peaks were between 0.6 and 0.9. When plotting all points with the value 0.6 in this case there was more than one focus spot. When visualizing all points with the value 0.9, however, there was a single focus spot. Simulations sorted into this category were considered reliable as well. The plots showing $\mathbf{I}_p$ in Fig. 1 would be sorted into category II because at 0.6 there are multiple black spots and at 0.9 there was only one focus visible.




Category III is depicted in Fig. 2 with two peaks above $0.9$. Both of these peaks are inside the source area. In general, a secondary focus introduces unwanted ambiguity to the localizations. It would be unclear which of the peaks is the real localization and which the artefact. In the cases we associated with category III, however, all foci were within the deviation threshold when viewing values of $0.9$. Therefore, we assumed they belong to the same localization. Depending on the type

of application, these localizations may be considered reliable or not. If an application includes only the localization of single events, these category III simulations can be used to localize the events. If, however, there is more than one event or the number of events is unknown, these simulations could be problematic.

Category IV simulations were regarded unsuccessful although the highest value was found to deviate less than the threshold value from the source. Focusing spots were outside of the source area and had a peak value of $0.9$ or higher. We assumed

that these artificial focusing spots appeared due to the model setup. Category IV simulation setups could potentially lead to erroneous localization results.

In summary, all simulations were first sorted into successful (close to source) and unsuccessful (far from source) localizations. Afterwards the successful localizations were sorted into the four described categories depending on their reliability. To determine which simulation setups will produce reliable localizations, the simulations with localizations in category I and II

should be viewed. Simulations with more than one imaging condition in category I or II were considered simulations producing reliable localizations. This requirement to have two successful imaging conditions per simulation inhibits the misinterpretation of an artefact for a localization. It could be crucial when using real data.

## 3 The influence of the station distribution

To discover prerequisites for localizing seismic sources with TRI, simulations were performed with varying station distribu-

tions. Whether the specific setup enhances localizations can be derived from the reliability of localizations achieved with that setup.

The reliability of localizations is influenced by the amount and position of stations at the surface. The wavefront is only sampled at these discrete positions. Therefore, different aspects of the station distribution were tested systematically to determine which station placements enable reliable and unambiguous localizations.

In the first set of simulations, the minimum aperture and maximum receiver distance were investigated. The ability to localize sources with an array not centred above the source was tested in a second set of simulations. In the third set of simulations, the ability of TRI to cope with heterogeneous station distributions was tested.

These three sets of simulations were performed with nine receivers at the top of the model. Additionally, the effect of using an increased number of receivers was investigated for individual setups. All simulations in this section were performed using

a homogeneous velocity model for forward as well as time-reversed simulations. The p-wave velocity was set to $4000\,\mathrm{ms}^{-1}$, the s-wave velocity to $2300\,\mathrm{ms}^{-1}$ and the density to $2000\,\mathrm{kgm}^{-3}$. There was no noise added to the traces. Therefore, the only change between simulations was the station distribution. Presumably, observed differences in localization quality were caused by the change in the station placement.



## 3.1 Minimum aperture and maximum inter-station distance

To find the minimum aperture and maximum inter-station distance still enabling localizations with TRI, nine stations were placed in a square and centred above the source location. The distance $d$ between the stations was then increased discretely (see Fig. 3). By increasing the inter-station distance, the aperture of the array is increased as well. When using nine stations in a square layout, the aperture is twice the distance $d$ between receivers. The expected result would be a range of receiver distances producing reliable localizations. The lower bound of this range corresponds to the minimum aperture and the upper bound to the maximum inter-station distance needed to localize a source of the specific type in the specific depth.

In Fig. 4 performed simulations are marked with a grey bar to distinguish gaps in simulations from simulations that did not localize the source successfully. Only successful localizations, deviating less than the threshold from the source location, are marked with a symbol corresponding to the imaging condition and a colour corresponding to the category. In all results in this study, $\mathbf{I}_s$ is excluded for explosion sources because the radiation pattern mainly produces p-waves and therefore does not allow any localizations with $\mathbf{I}_s$. Similarly, $\mathbf{I}_p$ is excluded for strike-slip sources. Three source depths were tested with an explosion and a strike-slip source. Inter-station distances producing localizations in category I or II for at least two imaging conditions were considered reliable. Category III localizations may introduce ambiguities.

Inter-station distances and subsequently apertures producing reliable localizations can be seen in Fig. 4. We were able to localize a strike-slip source in 5 km depth with nine receivers placed 1 to 9 km apart. The explosion source in 5 km depth was localized reliably with an inter-station distance of 6 km. For a 11.9 km deep source, a reliable localization was possible with inter-station distances of 4 to 19 km for a strike-slip source and 9 to 15 km for an explosion source. The strike-slip source in 22 km depth was localized with nine receivers spaced 17 to 25 km apart and the explosion source with receivers spaced 15 to 25 km apart. The strike-slip source in 22 km depth could also be localized with inter-station distances of 6 to 12 km. However, at an inter-station distance of 15 km there was only one imaging condition led to a reliable localization. Therefore only the larger inter-station distances were considered successful for this source depth.

In general, we found that sources in a certain depth could be localized with an inter-station distance roughly the same as the source depth and an aperture of twice the source depth. Additionally, the spread of distances that worked for a specific source depth increased with an increasing source depth. Furthermore, strike-slip sources could be localized with a wider range of distances than explosion sources.

In our setup with a homogeneous velocity model, nine receivers were sufficient to localize explosion and strike-slip sources reliably. In Fig. 5, the localization results for an increased number of receivers are shown. Receivers were added outside of the nine original receivers with the same inter-station distance. Receiver distances stayed constant (at 13 km for this source in 11.9 km depth) while the aperture was increased. Additionally, receivers were added inside the original array and consequently the aperture was kept constant at 24 km.

Increasing the number of receivers from 9 to 25 increased the quality of the localization. 49 receivers slightly enhanced the results further. Using 169 receivers, spaced 2 km apart, did not improve the localization quality anymore. For the strike-





slip source the localization quality even decreased for $\mathbf{I}_e$ to a category III localization. These results suggest that a slight improvement in localization quality can be achieved by using more stations. Nevertheless, nine stations produced reliable localizations with a homogeneous velocity model and no noise in the traces. The only simulations of this study where all imaging conditions produced a category I localization can be seen in Fig. 5 for the strike-slip source when using 25 or 49

stations. An explosion source could not be localized with three category I localizations.

In addition to the optimal range of inter-station distances, we found that different imaging conditions were sensitive to different source types. While $\mathbf{I}_d$ and $\mathbf{I}_s$ seemed to be successful for a strike-slip source, $\mathbf{I}_e$ and $\mathbf{I}_p$ were more successful for an explosion source. This can be seen especially well when considering only category I localizations in Fig. 4 and 5.

### 3.2    Maximum asymmetry over source location

To determine the sensitivity of TRI to an event not centred beneath the array, localizations were analysed with an array of nine receivers shifted discretely in one direction. In practice, arrays are rarely centred above an event location. Therefore, it is necessary to know in which area of the model sources can be localized. This helps to estimate either where to put stations if events are expected in a specific target area or where localizations can be expected when using an already existing array.

In this section, the asymmetry of the array in relation to the source location was investigated. Nine receivers were placed

in a square using the inter-station distances that produced reliable localizations found in the previous section. In Table 1, the used inter-station distance for each source depth can be seen. These nine receivers were moved in negative x direction in order to discretely increase the shift distance $s$ between the centre receiver and the source location (see Fig. 6). For smaller shift distances some part of the array is still above the source. At one shift $s$ the shift distance is equal to the inter-station distance. At this point one of the side receivers is directly above the source location. For larger shifts the source location is beside the

array. In Fig. 7 the shift distance $s$ where the source is no longer beneath the array is marked by a dashed line.

In Fig. 7 the results for different shift distances $s$ are shown. A $0\,\mathrm{km}$ shift represents an array that is centred above the source location. Shallow explosion sources, at $5\,\mathrm{km}$ depth, could not be localized with an array not centred above the source while strike-slip sources in $5\,\mathrm{km}$ depth could be localized with a shift $s$ of $6\,\mathrm{km}$. Reliable localizations were possible for the explosion source in $11.9\,\mathrm{km}$ depth at shifts of 6 and $8\,\mathrm{km}$. The strike-slip source in $11.9\,\mathrm{km}$ depth could be localized with shifts of 2,

13 and $19\,\mathrm{km}$. In-between those shift distances it can be either localized only with one imaging condition or not at all. The explosion source in $22\,\mathrm{km}$ could be localized with shifts from 8 to $13\,\mathrm{km}$ while the strike-slip source in this depth could be localized with shifts of $20\,\mathrm{km}$ and $23\,\mathrm{km}$.

Explosion sources could best be localized when the centre of the array was directly above the source or when the source was in-between the receivers. It could not be localized reliably if the source location was too close to one of the receivers or

beside the array. Strike-slip sources could be localized if one of the receivers of the array was above or near the source location but they could not be localized if the source location was between two stations. Neither the explosion source nor the strike-slip source could be localized when the source location was outside the array. The only exception was the strike-slip source in $11.9\,\mathrm{km}$ depth. It could be localized reliably with a shift of $19\,\mathrm{km}$.





To further investigate the different behaviour of the localization of explosion and strike-slip sources, simulations were performed with an increased amount of receivers. 25 receivers were placed in a square of 5 by 5. The inter-station distance was set to $8\,\mathrm{km}$ which is smaller than the inter-station distance used for the nine receivers in Fig. 7. An explosion and a strike-slip source in $11.9\,\mathrm{km}$ depth were tested. The results can be seen in Fig. 8. For both source types almost all tested shifts of the array

were able to localize the sources reliably. There is no significant difference between simulations where one of the receivers was above the source location ($s = 8\,\mathrm{km}$ and $s = 16\,\mathrm{km}$) and those where the source location was in-between the receivers ($s = 4\,\mathrm{km}$ and $s = 12\,\mathrm{km}$). From a shift distance of $16\,\mathrm{km}$ onwards, the source was beside the array. Both source types could be localized reliably when the source was next to the array up to a shift of $32\,\mathrm{km}$ where the source was $16\,\mathrm{km}$ next to the array. At a shift of $24\,\mathrm{km}$, the explosion source could not be localized reliably anymore while at a shift of $28\,\mathrm{km}$ the strike-slip

source could not be localized reliably anymore. An overall decrease in localization quality was observed when the source was no longer beneath the array.

Overall, more receivers spaced closer together than suggested from the previous section reduced the effect of source types not being localized beneath the whole array. When the source was outside of the array, localizations became impossible when using only nine receivers. When using 25 receivers, localizations were possible also outside the array.

## 3.3 Maximum heterogeneity of azimuthal gaps between stations

To determine the influence of heterogeneous inter-station distances on the localization capabilities of TRI, the azimuthal gap $a$ was used. Seismic stations are rarely placed in a square with every receiver the same distance away from the next. Horstmann et al. (2015) introduced the azimuthal gap as the horizontal angle between two stations viewed from the projection of the source to the surface. With nine receivers placed in a square, the azimuthal gap is $45°$ for each station pair, excluding the centre

station. When moving the corner stations closer to the source location in x direction, the angle increases between some of the stations and decrease between other stations (see Fig. 9). In this study, the azimuthal gap refers to the maximum angle between stations. The minimum azimuthal gap can be derived by subtracting the maximum azimuthal gap from $90°$.

In Fig. 10 the localization results are shown. Because the stations could only be moved on the model grid, the resulting azimuthal gaps are spaced differently for the three source depths. We found that the deeper the source the more heterogeneous

the stations could be distributed. For a $22\,\mathrm{km}$ deep source, reliable localizations were possible with a maximum azimuthal gap of up to $67°$. A $11.9\,\mathrm{km}$ deep source could be localized reliably with an azimuthal gap of up to $58°$. The shallow source, in $5\,\mathrm{km}$ depth, did not seem to allow a heterogeneous station distribution.

## 4 The effect of a complex velocity model

In addition to the station distribution, a complex velocity model may have a strong influence on the localization capabilities

of TRI. In general, a sufficiently correct velocity model is needed to be able to backpropagate the wavefield adequately. But also a sufficiently correct velocity model may include complex velocity structures such as low velocity zones or sharp velocity





contrasts. These structures can trap energy and obscure the source localization. This is why the imaging conditions were divided by an illumination map as described in section 2. However, pronounced effects may remain in the final result.

To investigate the extend of focusing errors due to complex velocity structures, four different velocity models were tested. Differently shaped and positioned low velocity zones were tested as well as a fault. Each velocity model was once tested with

the low velocity zone or fault known and once with the low velocity zone or fault not known. The velocities in the low velocity zones were $75\%$ of the values from the homogeneous models ($v_p = 4000\,\mathrm{m/s}$). The layered model used four layers and p-wave velocities ranged from $3000\,\mathrm{km}$ to $6000\,\mathrm{km}$. S-wave velocities were calculated with a constant ratio of $v_s = v_p/\sqrt{3}$.

The forward simulation incorporated the structure while one backward propagation was done with the exact same velocity model as the forward simulation and one was done with a homogeneous or layered velocity model instead. For each velocity

model, a strike-slip source and an explosion source was tested in $11.9\,\mathrm{km}$ depth. Two sets of receivers, one with nine receivers and one with 25 receivers, were placed in a square and centred above the source. The receivers were spaced $13\,\mathrm{km}$ apart for both sets.

In Fig. 11 the localization results are shown. The results of the two sets of receivers used are shown in the upper and middle part of the plot. The velocity model used for the forward simulation and one of the time-reversed simulations is shown in the

lower part of the plot. For the second time-reversed simulation a homogeneous model or a simple layered model was used.

When there is a known low velocity layer above the source (see Fig. 11, left model), both a strike-slip source and an explosion source could be localized reliably. There is no drastic improvement when using more receivers. When the low velocity layer above the source was not known, the localization capabilities decreased and a reliable localization was no longer possible.

When the source was inside the low velocity layer (see Fig. 11, second model), the localization was not significantly en-

hanced by using more receivers as well. When this low velocity zone was not known, the reliable localization of a strike-slip source was still possible but not the localization of an explosion source.

When there was a spherically shaped low velocity zone around the source (see Fig. 11, second model from the right), the localization was enhanced when using 25 receivers. A strike-slip source inside a spherically shaped low velocity zone could be localized when the low velocity zone was known but not at all when it was not known while an explosion source could still be

localized when the low velocity zone was not known.

The last velocity model was a layered model with a fault (see Fig. 11, right model). When the fault was known the strike-slip source could be localized with 9 and 25 receivers. When it was not known, when just a layered velocity model was used for the backward propagation, the reliability decreased and the source could not be localized anymore. An explosion source could not be localized successfully, even with 25 receivers.

Taken together these results suggest that an increased number of receivers helps only slightly when the velocity model was more complex. Because the additional receivers were placed the same distance apart from the original nine receivers, the main characteristic helping the localization was an increased aperture. Additionally, the localization of strike-slip sources was influenced less by a complex velocity model than explosion sources. If structures like low velocity zones and faults were known, it did not hinder localizations. However, if they are not known, localization quality decreased.



## 5 Limits for the signal-to-noise ratio

The previous sections used only synthetic seismograms recorded during forward simulations and then re-emitted into the model time-reversed. Real data, however, does not only consist of the signal corresponding to the event. It also includes a variable amount of noise. In theory, TRI works with noisy data (Gajewski and Tessmer, 2005; Witten and Artman, 2011). In this section, the smallest signal-to-noise ratio (SNR) still enabling localizations was investigated. An explosion source and a strike-slip source in $11.9\,\mathrm{km}$ depth were used. 9 and 25 receivers were spaced $13\,\mathrm{km}$ apart and centred above the source. Five different SNR values were used: 1, 0.75, 0.5, 0.25 and 0.1.

The noisy data was created by constructing a time series with random amplitudes and the same time step and length as the seismograms. Afterwards the noise signal was filtered to exclude numerically unstable frequencies above $5\,\mathrm{Hz}$. The amplitudes of this time series were modified to achieve the desired SNR of this noise signal in relation to the seismograms of all stations. The filtered noise signal was afterwards added to the seismograms of all stations. Figure 12 shows example traces of the seismograms without noise and the seismograms with the respective SNR value for a receiver directly above an explosion source. When the SNR decreases, the event signal becomes less and less distinguishable from the noise.

Figure 13 shows the localization results. When using nine receivers there was no reliable localization possible for both source types. With a SNR of 1 there was only one localization in category II and for SNRs down to 0.5 there were only localizations in category III. When using 25 receivers it was possible to localize both source types with a SNR as low as 0.25 in category I but only with one imaging condition. The explosion source could be localized in category I with two different imaging conditions with a SNR as low as 0.5. Surprisingly, the total energy density imaging condition $\mathbf{I}_e$ seemed to be the most stable one for localizing sources of both source types. In the previous sections, $\mathbf{I}_e$ seemed to be not as suitable for a strike-slip source.

Overall these results suggest that the noise level has a higher impact on localization quality compared to complex velocity models. It was possible to localize sources even if the noise level was too high to distinguish events in the traces. Nevertheless very low SNR hinder a reliable localization which can only partly be compensated by adding more receivers.

## 6 Application example for Southern California

In previous sections, station distributions were tested systematically to gain insight into the characteristics of array designs that influence localization results. Additionally, results showed that noisy data influence the localization more than a complex velocity model if the velocity model is accurate enough. In this section, an example of applying these results is shown. First, the given stations are evaluated for their ability to localize events with a homogeneous velocity model. We show that subsets of the array may enhance localization quality and how to design an array in this region. In the second part, we estimate the success rate achievable with different station distributions using the velocity model by Zeng et al. (2016) and noisy data.





## 6.1 Determine localization possibilities using results from previous sections

Horstmann et al. (2015) localize non-volcanic tremor in Southern California near Cholame and use 39 stations. In this study, we took 38 of the 39 stations (the northernmost station was excluded here) and tried to localize synthetic events. The receiver positions were transformed to the same x-y-grid as in Horstmann et al. (2015) and plotted in Fig.(a). Additionally, three source

positions are plotted. These source positions will be used in the following to test the possibility to localize events with one of three receiver sets derived from the 38 stations in Southern California and one receiver set suggested as an optimal array following the results of the previous sections. As source type, strike-slip sources were used in this section. Strike-slip sources occur more dominantly in the subsurface than explosion sources. The extension of this model in x and y direction is the same as in previous models as well as in Horstmann et al. (2015).

Figure 14(a) shows that the 38 stations are positioned heterogeneously and extend farther in y direction than in x direction. Because the receivers are so heterogeneously spaced, this set of receivers was not expected to give a reliable localization for any of the three sources. The total amount of stations, however, may counteract the heterogeneous inter-station distances. Additionally, we determined two subsets of receivers with more homogeneous inter-station distances. Two subsets were chosen here: one with 31 stations (see Fig. 14(b)) and one with 20 stations (see Fig.14(c)). For receiver set (b) stations very close to

each other were excluded while for set (c) stations were reduced to decrease the total amount of traces, which may impact computation time. Lastly, in Fig. 14(d) an optimal array is suggested. This array was designed with a homogeneous station distance of $8\,km$ and an aperture of $24\,km$. The array was positioned to allow localizations in roughly the same areas as in Fig. 14(c). For a real application, the array would be moved to localize events in the specific target area. The inter-station distances and apertures for the arrays can be found in Table 2.

The target depths for localizations was $10\,km$ to $25\,km$ because bigger events occur at depths as shallow as $10\,km$ while the non-volcanic tremor signals seem to occur deeper (Horstmann et al., 2015). For this target depth range, the results of section 3.1 suggested the aperture should be about $20\,km$ or larger to be able to localize deeper events and the receiver distance should not be larger than about $13\,km$ to include shallower events. When comparing these requirements to the values reported in Table 2, the minimum aperture was greater than $20\,km$ for all four receiver sets. Similarly, the average inter-station distance was

smaller than $13\,km$ for all four receiver sets. However, the inter-station distances vary greatly, especially for receiver set (a) and (b).

In Fig. 14 two circles mark areas where less or more reliable localizations are expected for the target depth range. These circles were derived from the results from previous sections where we discovered that sources could be localized beneath the array when using fewer stations and slightly outside the array when using an increased amount of receivers. Three source

positions were chosen to test the hypotheses of where localizations were expected.

Simulations were performed with a strike-slip source and a homogeneous velocity model (see top of Fig. 15). The source was placed at all three source positions and in two depths. The four receiver sets presented in Fig. 14 were tested for their capability to localize these sources.





Localizations were possible with receiver set (a) although inter-station distances were very heterogeneous. At source position 1, a reliable localization was possible in a depth of $11.9\,\mathrm{km}$. In $22\,\mathrm{km}$, a reliable localization was not possible. At source position 2 and 3 a localization in $22\,\mathrm{km}$ was possible.

Receiver set (b) has a slightly enhanced localization quality compared to set (a). All sources could be localized reliably. This

highlights that TRI is rather stable and only small alterations to the station distribution allow a localization of events in multiple depths at different positions in the model.

Receiver set (c) consisted of 20 stations and localization quality was decreased for the sources in $22\,\mathrm{km}$ depth. An explanation for the decreased quality for deeper sources could be the reduced aperture of this receiver set. The source in $11.9\,\mathrm{km}$ depth could, however, be localized reliably and with all three imaging conditions in category I or II. This is coherent with previous

results suggesting a decrease of sensitivity to the station distribution with depth.

Receiver set (d) was suggested as an optimal array based on the results of the previous sections. This was expected to be the most successful receiver set because the inter-station distance was chosen to fit the target depths and was homogeneous and the aperture was chosen to be wide enough for deeper sources. The source in $11.9\,\mathrm{km}$ depth could, however, not be localized reliably. Only one imaging condition produced a localization in category I while the other two were in category III.

This localization result was, therefore, less reliable than that of receiver set (c). However, both receiver sets have the same amount of stations and the aperture is larger for receiver set (d) than for (c). The source is also beneath the array in both cases. In receiver set (d), the distance between receivers is $8\,\mathrm{km}$. In section 3.1 a strike-slip source at the same location could be localized with nine receivers $8\,\mathrm{km}$ apart with $\mathbf{I}_e$ and $\mathbf{I}_s$ in category I quality. This suggests that additional stations that add to the asymmetry of the array (from 3 by 3 stations to 4 by 5 stations) produced a split of the focus. However, receiver set (b) and

(c) were asymmetric as well as they are elongated in y direction. An alternative explanation could be that too regular positioned stations actually slightly decrease the localization quality. The localization quality with receiver set (d) for $22\,\mathrm{km}$ deep sources was similar to the results with receiver set (c) suggesting that in larger depths the unknown effect is negated.

After performing this synthetic study with a homogeneous velocity model, we would decide to use receiver set (b) for localizing events in this region if these are the only available stations. If planning to deploy stations, receiver set (d) needs to

be further adjusted to increase the localization quality.

## 6.2  Synthetic tests to estimate success rate with real data

After finding a suitable receiver set enabling the localization with a homogeneous velocity model in different source depths and at different source positions, the success rate with the real velocity model and noisy data has to be investigated. Therefore, additional simulations were performed.

Zeng et al. (2016) obtain the most current and accurate velocity model for the region in Southern California in which Horstmann et al. (2015) locate non-volcanic tremor. The velocity model is three-dimensional and incorporates p- and s-wave velocities. It was interpolated to fit the used finite difference grid and the density was calculated from the p-wave velocity with





the empirical equation by Brocher (2005):

$$\rho = 1.6612 v_p - 0.4721 v_p^2 + 0.0671 v_p^3 - $$
$$0.0043 v_p^4 + 0.000106 v_p^5. \tag{6}$$

$v_p$ depicts the p-wave velocity in $\mathrm{kms^{-1}}$ and the resulting density $\rho$ is in $\mathrm{gcm^{-3}}$.

In Fig. 15 at the bottom, localization results for simulations with and without added noise are shown. Without any noise, the strike-slip source at position number 1 could be localized reliably in $11.9\,\mathrm{km}$ depth with receiver set (b) and in $22\,\mathrm{km}$ depth with receiver set (a). Receiver set (c) was not able to produce any localization with the actual velocity model. Receiver set (d) localized sources in both depth in category I with $\mathbf{I}_e$ but only in category III for the other imaging conditions.

    When adding noise with a SNR of 1, reliable localizations were possible in both depths with receiver sets (a) and (b) and

additionally for the deeper source with receiver set (d). Receiver set (c) was still unable to localize any of the sources.

    Receiver set (b) remained the best choice in this case followed by receiver set (a). Both were able to localize three out of four sources. Both sets localized the sources with noise reliably. Surprisingly, receiver set (d) was able to localize the deep source with added noise reliably but not without any noise. This adds to the theory that a very regular station distribution may even hinder localizations. Noise has in this case a similar effect on the wavefield as irregular inter-station distances.

we would suggest to use receiver set (b) to localize events in this area and avoid ambiguity, because all localizations were in category II and there was no localization in category III. The success rate of localizations in this case is expected to be high inside the grey circles marked in Fig.14 as long as the noise level is not too high. Localization quality is expected to decreases with increasing distance from the array.

## 7   Discussion

The objective of this study was to find station distributions producing reliable localizations with TRI. Additionally, the knowledge about the influence of a complex velocity model and a low signal-to-noise ratio on the localization quality enables the estimation of localization quality achievable with real data. This is important for (1) estimating the success rate of TRI with a given array and velocity model, (2) decreasing the time needed to adjust the stations (e.g. by choosing only a subset of receivers) and (3) designing an array for localizing events in a designated target area.

The method of TRI is straight-forward and can be to implemented into most numerical codes that can propagate elastic waves through a medium. In this study, we further improved the workflow of Saenger (2011) which was modified by Witten and Artman (2011). We suggested to visualize all points with certain fractions of the highest value in imaging conditions and proposed a set of categories to differentiate between localization qualities. The purpose of this workflow was to assess which simulations gave the most reliable localization. When using real data, the sorting into categories may be obsolete. If the receiver

distribution is sufficiently well and the level of noise low enough, all focusing spots appearing in the imaging conditions should be localizations.



Additionally, we rated the success of localizations by the deviation of the maximum value in the imaging conditions to the initial source location. The used threshold deviation was calculated as the average deviation from all three directions. Therefore, successful localizations for the shallowest source could be at the surface. Surface localizations were, however, excluded. This restricted the range for a successful localization for the source in $5\,\mathrm{km}$ depth. Consequently, it looked like a source in $5\,\mathrm{km}$

depth was especially hard to localize. However, sources shallower than $10\,\mathrm{km}$ are rarely observed in the region in Southern California (Horstmann et al., 2015).

All simulations in this study were set up with the same frequency range for the source wavelet, the same grid spacing, and the same time step. For explosion sources this setup was numerically stable in the used range of seismic velocities. However, S-waves and surface waves experienced numerical dispersion. This could not be eliminated without reducing the grid spacing

and consequently increasing computation time significantly. Therefore, we kept the chosen values. We observed no effect in the results suggesting that dispersive s-waves interfered with the localization. The surface waves were more dispersive than the s-waves but are not relevant for this method and consequently do not interfere with localizations as well. The dispersive surface waves may be an alternative reason why we needed to exclude focusing at the surface.

### 7.1  Station distribution for reliable localizations

Numerous simulations were performed to test different station distributions for their ability to localize sources of different source types as well as different depths. We found that inter-station distances should not be larger than the source is deep. Additionally, the total aperture of the array should be at least twice the source depth.

Lokmer et al. (2009) observed that inter-station distances of up to half of the dominant p-wave wavelength are still able to localize events. As we performed all simulations with the same frequency range for the source wavelet in the forward

simulation. We were unable to judge the influence of varying wavelengths on localization results. However, the localization quality changed with different source depths and inter-station distances. Additionally, inter-station distances found to produce reliable localizations were significantly larger than half the dominant wavelength, which is $1.2\,\mathrm{km}$ in our case.

Furthermore, Witten and Artman (2011) stated that the backpropagated wavelength is undersampled because traces are only inserted at discrete positions at the surface. The depth in which the wavefront is healed and thus a localization can occur seems

to depend on the distance between the stations. They observed that at a depth of 1 to 1.5 times the inter-station distance the wavefront is healed, which means stations should be spaced at a distance equal to the source depth or slightly closer. This is coherent with our results.

A common prerequisite when applying TRI is a sufficiently large aperture. This is consistent throughout the literature. Artman et al. (2010) observe a greater influence of the aperture on the horizontal than the vertical resolution. We could not observe

this effect in our study but agree that a large aperture is needed for reliable localizations.

The total number of stations is often considered a crucial factor controlling localization quality. We found that in the simple case of a homogeneous velocity model and noise-free data, nine stations were sufficient to localize events reliably, even if they are spaced slightly heterogeneously. More stations increased the quality of the localization slightly. However, a significantly





large number of stations did not produce perfect localizations but in some cases even decreased localization quality. Additionally, we found a slightly increased amount of stations may help counteract effects of a complex velocity model and high noise in the data. Horstmann et al. (2015) suspect that the distribution of stations may be more influential than the total number of stations.

No study reported how far outside of the array sources can be localized. However, this is crucial when designing an array because the array should be positioned so events can be localized in the target region. We found that a smaller amount of stations enhances localizations beneath the array. When reducing the inter-station distance and increasing the amount of stations, events could be localized outside the array as well. The reliability of localization, however, decreased when sources were outside of the array.

Horstmann et al. (2015) observed that the maximum azimuthal gap, which is a measurement for the heterogeneity of inter-station distances, influences the localizations. We observed a similar behaviour and additionally found that with increasing source depth, the allowed heterogeneity increases. The azimuthal gaps reported by Horstmann et al. (2015) are, however, at least twice what we found to be the maximum azimuthal gap. In our case with nine stations, an increase in the maximum azimuthal gap between two stations subsequently resulted in a decrease in the minimum azimuthal gap between two other

stations. The discrepancy with Horstmann et al. (2015) suggests that the minimum azimuthal gap (how close two stations are together) has a higher influence than the maximum azimuthal gap on the localization results. This is also supported by the improvement in localization quality when using 31 of the 38 stations in Southern California. Stations close together were excluded to form a subset of the 38 stations. On the other hand, some studies (e.g. Larmat et al. (2009) or Montagner et al. (2012)) compensate an irregular station distribution by weighing the stations according to an area assigned to each station. Localization

results seem to improve when using this method. Our results suggest, however, that by omitting a few stations results can be improved significantly. Consequently, the weighing of the stations may become obsolete. When the aim is to have a fast and reliable localization method, the gain in localization quality has to be balanced with the extra computation time that is needed for weighing the stations.

While investigating optimal inter-station distances, we observed that strike-slip sources could be localized with a much wider range of inter-station distances and apertures as explosion sources. We also observed that strike-slip sources could be localized most reliably when one of the receivers was above the source and explosion sources could be localized most reliably when the source was in-between stations. We therefore conclude that for strike-slip sources the receiver directly above the source has a greater influence on the localization result and for explosion sources the receivers farther away have a greater

influence. Explosion sources are therefore more sensitive to the positioning and number of receivers while strike-slip sources are more sensitive to how close a receiver is. This could also explain the observation of Steiner et al. (2008) that the most accurate localizations can be gained when using sources emitting strong s-waves in a vertical direction, as is the case with some strike-slip sources. It suggests their aperture was not wide enough or their inter-station distance too large for localizing sources emitting mainly p-waves.





In the used workflow, we set the first $2.3\,\mathrm{km}$ in the imaging conditions to zero to remove any focusing at the surface. This could be extended further to reduce the area where focusing is allowed to the area underneath the receivers. However, we observed that sources could be localized outside the array as well if there were enough stations. The transition between areas where localizations were possible and impossible was not a clear line but a transitional zone. Additionally, depending on the

velocity model and the position of the source in relation to velocity structures, waves could be scattered and sources could be localized outside of this region as well. In contrast to reducing the area where focusing is allowed, we propose to carefully evaluate focusing spots outside of the array. Furthermore, if a focusing spot is outside the array but the area where focusing is allowed is reduced to the area beneath the array, normalisation of imaging conditions leads to an amplification of amplitudes inside this area beneath the array. By construct, there will be one point in this area with the highest value of 1 and there will

be spots of focusing. Whereas, when allowing localizations in the whole model, the location of the highest value may indicate that it is an artificial focusing spot and additionally, amplitudes in the area are not artificially increased. This allows for the possibility that there is no focusing spot beneath the array.

## 7.2 Evaluation of success rate with real data

Preliminary synthetic studies are typically used to evaluate the localization quality with TRI achievable with a certain station

distribution. Often the first simulations are performed with a homogeneous velocity model. If these simulation are successful and a station distribution yields reliable results, the real velocity model is used. If that is successful again, the real data is used that contains varying amounts of noise.

We showed that complex velocity structures do not hinder localizations with TRI if they are known. Low velocity zones did not prevent reliable localizations even when using only nine stations. If the low velocity zone was not known, localization

quality decreased. Additionally, we saw a similar effect for a layered velocity model with a fault. The velocity model can be fairly complex and does not interfere with the localization if the velocity structures are sufficiently known. Additionally, we observed that noisy data can hinder localizations. When using noisy data, more receivers were helpful to still be able localize sources. However, when adding noise to the traces, the same noise signal is added to all stations. The noise is still random but the same for all stations. This could unintentionally result in a structured noise that is not random anymore. Nevertheless, when

this noisy signal is inserted into the model it will travel in different directions and will interfere differently with each other, thus creating a noisy wavefield.

Furthermore, in some cases we observed an increase in localization quality when using noisy data compared to simulations without noise. Additionally, we observed an increase in localizations quality when using 20 heterogeneously spaced stations compared to 20 homogeneously spaced station when using a homogeneous velocity model. That suggests that in some cases

a very regular station distribution or traces with no noise may actually hinder localizations. It also stresses the necessity to attempt to localize a real event with the given array and velocity model before abandoning the study.

The scope of this study was limited in terms of computation time and therefore only distinct aspects of the station distribution were investigated. One additional aspect that could influence the localization quality are the influence of an asymmetric array.




An asymmetric array that extends further in one direction than in the other may decrease localization quality. We observed, that it may cause the focus to be split into multiple focusing spots and therefore introducing ambiguity. However, the study should give fundamental guidelines into the placement of stations for localizing sources with TRI.

TRI can be used for different scales as well as different applications. In this study we concentrated on the field scale. However, it would be a valuable addition to this study if a similar study would be performed on different scales to see if the same guidelines stay true on other scales as well. If the same general principles hold true for smaller or larger scales, it would suggest that observed effects are inherent of the method.

In this study, the event time was known and run-times were adjusted accordingly. When using real data, the event time is not known and therefore a strategy has to be developed to find accurate event times as well. Possibly, the station distribution, the velocity model and the noise also have an influence on the event time. This has to be investigated in future studies.

Lastly, we observed that different imaging conditions were sensitive to different source types. This could be further investigated to potentially derive the source type from results already obtained while localizing the source.

## 8 Conclusions

This study aimed at investigating the prerequisites for applying TRI on the field scale. We, therefore, systematically tested different station distributions with a homogeneous velocity model and evaluated the resulting localizations for their reliability. We found that the inter-station distance should be not larger than the source depth and the aperture of the array should be at least twice the source depth. Additionally, sources could be localized best when they were underneath the array. When using more stations, localizations outside of the array became possible. Stations should be spaced homogeneously but the deeper the source the more heterogeneously the stations could be.

Complex velocity structures did not hinder localizations if they were included in the velocity model while noise could inhibit reliable localizations, especially when using too few stations. The localization quality is, however, increased when using a complex velocity model with noisy data or a homogeneous velocity model with heterogeneous inter-station distances. This suggests that even when synthetic studies fail, localizations may still be possible with the real data.

We showed that only a few simulations are necessary when performing a preliminary synthetic study to estimate if the given setup will work for TRI. Additionally, we showed that for designing an array that should be used with TRI the target area should be considered. The depth of expected events then dictates the receiver distance and aperture of the array. If considering a range of depths, the inter-station distance should fit the shallowest events and the aperture the deeper events.

*Competing interests.* The authors declare that they have no competing interests.



*Acknowledgements.* This project has received funding from the European Union's Horizon 2020 research and innovation programme under grant agreement No. 727550 as well as the "Landesprogramm für Geschlechtergerechte Hochschule des Landes NRW" from the Ministry of Innovation, Science and Research of North Rhine-Westphalia, Germany. The authors gratefully acknowledge the computing time granted by the John von Neumann Institute for Computing (NIC) and provided on the supercomputer JURECA at Jülich Supercomputing Centre (JSC), Germany.



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





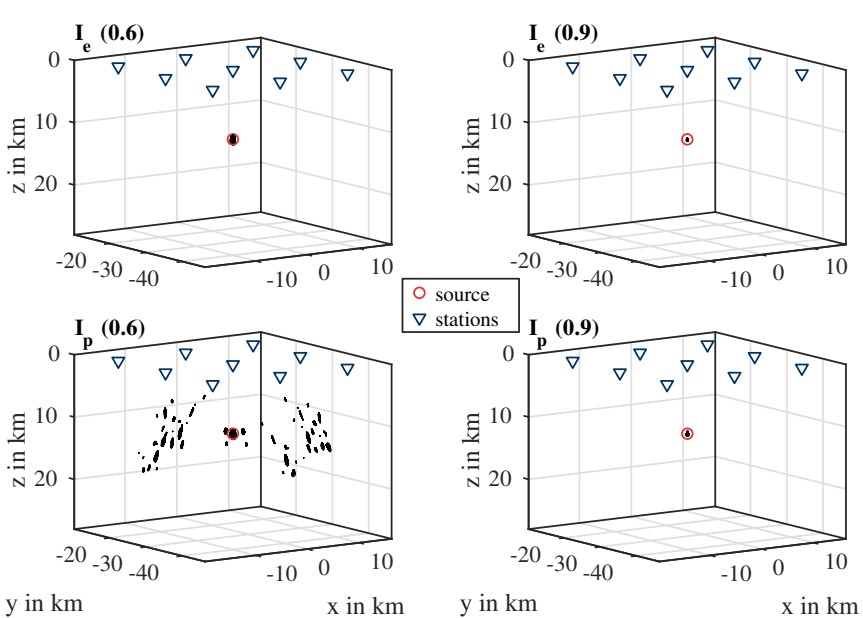

**Figure 1.** Imaging Conditions plotted for one example simulation. Black spots are points having the plotted value (0.6 on the left and 0.9 on the right). $\mathbf{I}_e$ would be classified into category I and $\mathbf{I}_p$ would be classified into category II. This example shows an explosion source in 11.9 km depth with 9 stations placed in a square. The inter-station distance is 13 km. Only part of the model is shown.





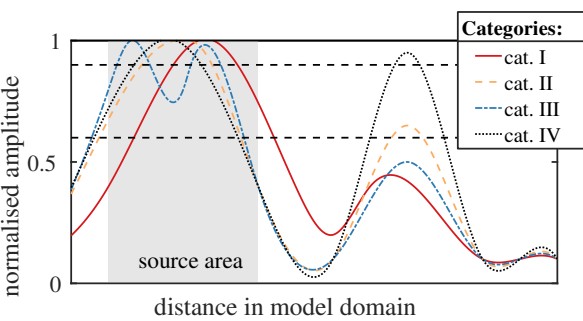

**Figure 2.** One-dimensional plot outlining four different categories that were used to rank simulation setups by their localization capabilities.





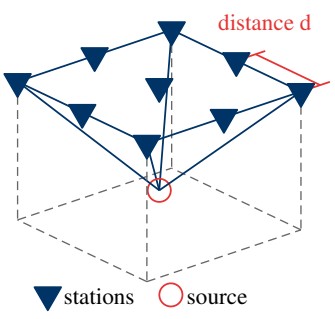

**Figure 3.** Placement of the nine stations used as receivers for testing the influence of the inter-station distance $d$. The middle receiver is directly above the source.





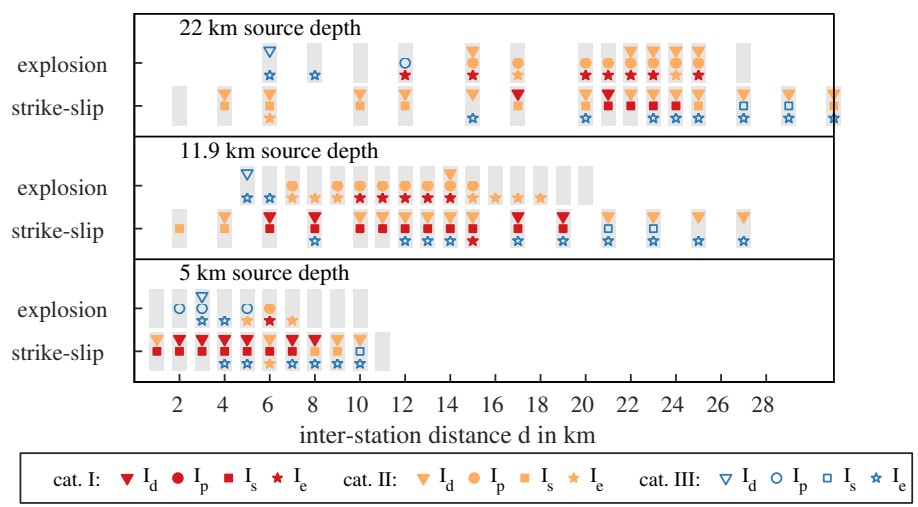

**Figure 4.** The quality of localizations achieved with different inter-station distances $d$. Three source depths were tested for two source types each. The localization achieved with the individual imaging conditions were ranked according to the categories from Fig. 2. Grey bars indicate setups that were tested. No symbol in the grey bar means the localization was not successful. Symbol type and colour in the grey bar represent successful localizations with the imaging condition sorted into the respective category.



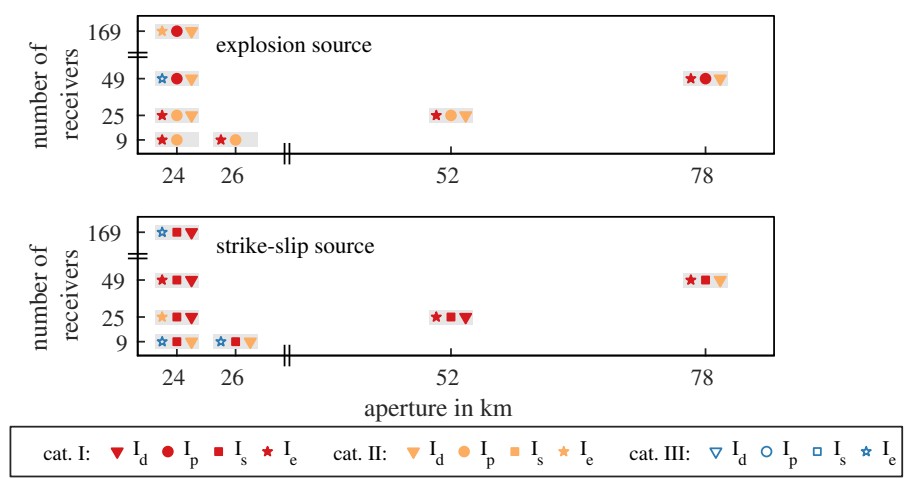

**Figure 5.** The quality of localizations achieved with an increased number of stations. Two source types were tested in $11.9\,km$ depth. Receivers were added either in-between the existing nine stations (aperture stays the same) or outside of the existing nine receivers (aperture increases). The localization achieved with the individual imaging conditions were ranked according to the categories from Fig. 2. Grey bars indicate setups that were tested. No symbol in the grey bar means the localization was not successful. Symbol type and colour in the grey bar represent successful localizations with the imaging condition sorted into the respective category.

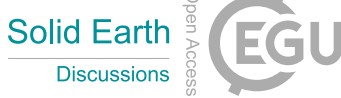

**Table 1.** Receiver distances used for different source depths according to results from section 3.1

| source depth (km) | receiver distance d (km) |
|:---:|:---:|
| 5 | 6 |
| 11.9 | 13 |
| 22 | 23 |





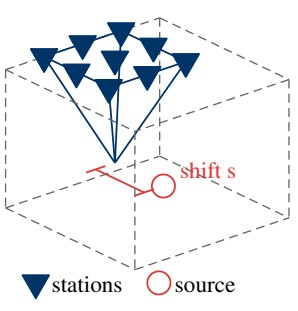

**Figure 6.** Placement of the nine stations used as receivers for testing the influence of an array that is not centred above the source. The shift $s$ was discretely increased.



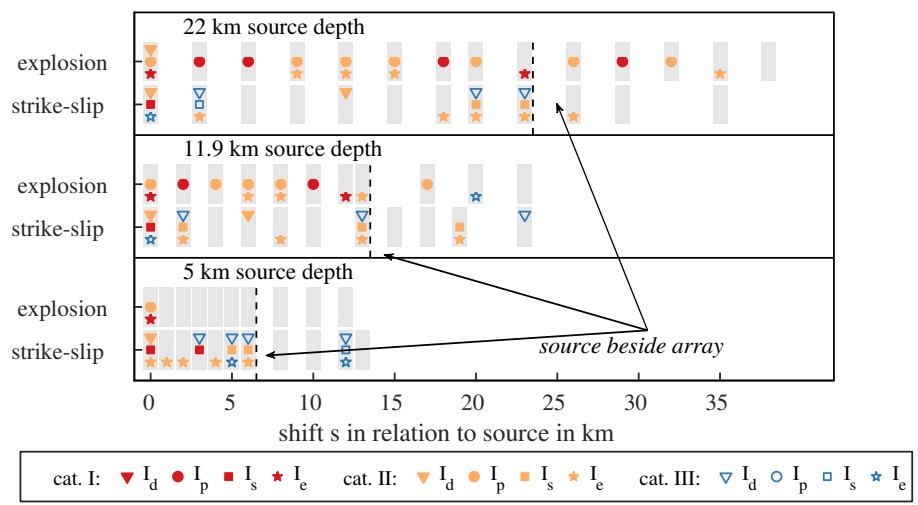

**Figure 7.** The quality of localizations achieved with different shift distances $s$ between the middle of the array and the source location. It is marked at which shifts the array is no longer above the array. Three source depths were tested for two source types each. The localization achieved with the individual imaging conditions were ranked according to the categories from Fig. 2. Grey bars indicate setups that were tested. No symbol in the grey bar means the localization was not successful. Symbol type and colour in the grey bar represent successful localizations with the imaging condition sorted into the respective category.





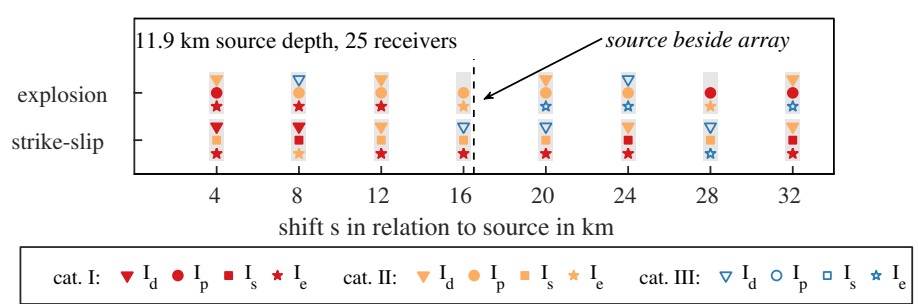

**Figure 8.** The quality of localizations achieved by using 25 instead of 9 stations placed $8\,km$ apart while shifting the array . It is marked at which shifts the array is no longer above the array. A source in $11.9\,km$ depth was tested with two source types. The localization achieved with the individual imaging conditions were ranked according to the categories from Fig. 2. Grey bars indicate setups that were tested. No symbol in the grey bar means the localization was not successful. Symbol type and colour in the grey bar represent successful localizations with the imaging condition sorted into the respective category.



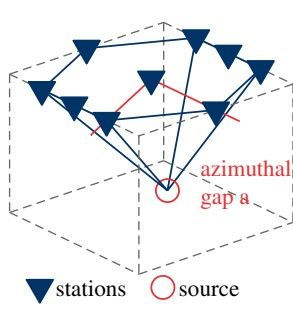

**Figure 9.** Placement of the nine stations used as receivers for testing the influence of an stations that are distributed heterogeneously. The maximum azimuthal gap $a$ is the largest found angle between two station viewed from the centre station. The azimuthal gap was increased by moving the corner stations closer toward the side receivers on two sides.





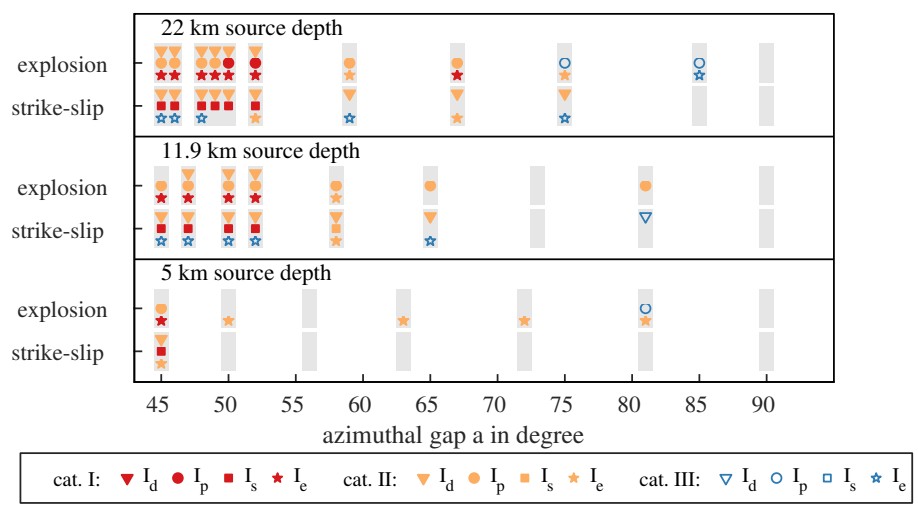

**Figure 10.** The quality of localizations achieved with different maximum azimuthal gaps $a$. Three source depths were tested for two source types each. The localization achieved with the individual imaging conditions were ranked according to the categories from Fig. 2. Grey bars indicate setups that were tested. No symbol in the grey bar means the localization was not successful. Symbol type and colour in the grey bar represent successful localizations with the imaging condition sorted into the respective category.



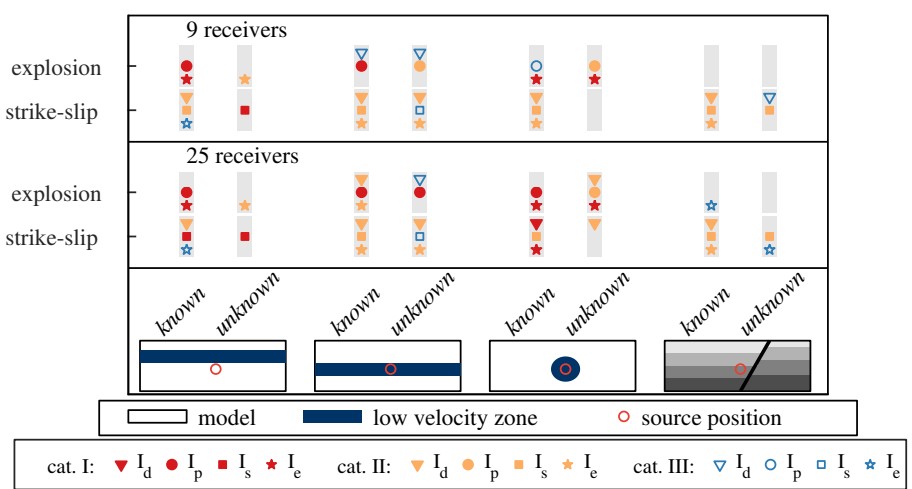

**Figure 11.** The quality of localizations achieved with different velocity models. For each velocity model one simulation was performed with the low velocity zone or fault known and one with a homogeneous or layered velocity model for the backward simulation. A set of nine receivers and a set with 25 receivers was tested. The localization achieved with the individual imaging conditions were ranked according to the categories from Fig. 2. Grey bars indicate setups that were tested. No symbol in the grey bar means the localization was not successful. Symbol type and colour in the grey bar represent successful localizations with the imaging condition sorted into the respective category.



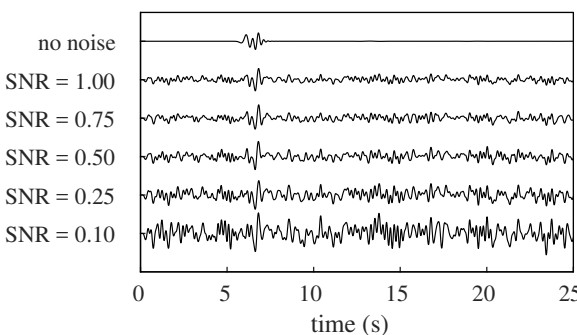

**Figure 12.** One example trace recorded with a receiver directly above the source. The explosion source was placed in $11.9\,km$ depth. The trace without noise and with different degrees of noise is shown, resulting in different signal-to-noise ratios (SNR).



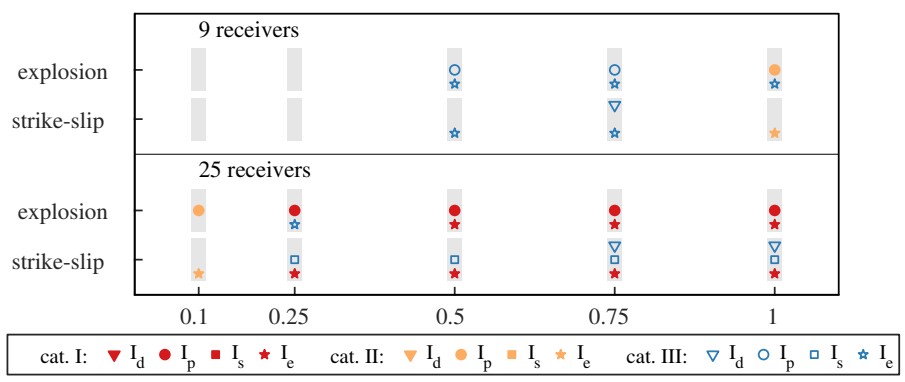

**Figure 13.** The quality of localizations achieved with different amounts of noise added to the traces. A set of nine receivers and a set with 25 receivers was tested for two different source types. The localization achieved with the individual imaging conditions were ranked according to the categories from Fig. 2. Grey bars indicate setups that were tested. No symbol in the grey bar means the localization was not successful. Symbol type and colour in the grey bar represent successful localizations with the imaging condition sorted into the respective category.





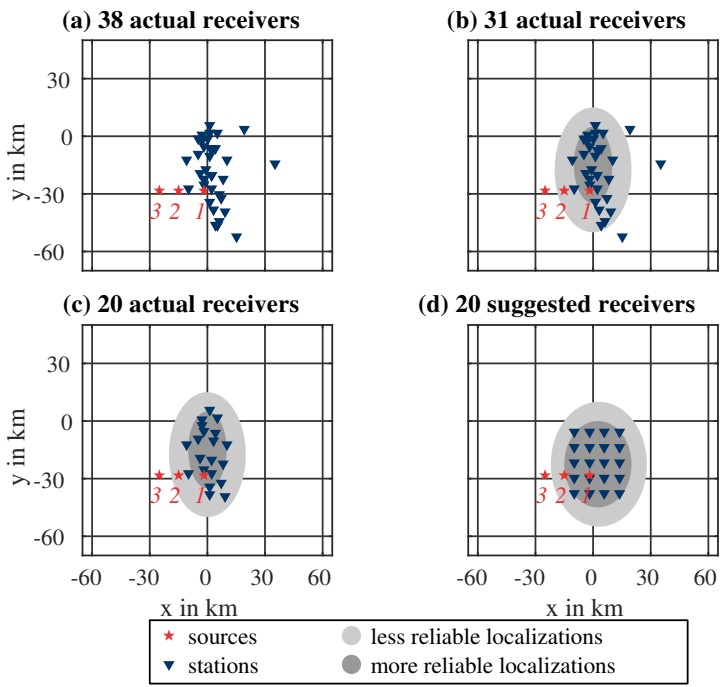

**Figure 14.** Receiver sets used to test success rate of localizations in Southern California: (a) 38 stations as used by Horstmann et al. (2015), (b) a subset of (a) excluding stations that are very close to another station, (c) 20 stations of (a) and (d) 20 stations suggested for optimal results. Three source positions are chosen to test the range of localization.





**Table 2.** Inter-station distances $d$ and apertures $a$ of the four receiver sets shown in Fig. 14

| receiver set (from Fig. 14) | number of stations | min $d$ (km) | max $d$ (km) | mean $d$ (km) | min $a$ (km) | max $a$ (km) |
|:---:|:---:|:---:|:---:|:---:|:---:|:---:|
| (a) | 38 | 0.138 | 23.94 | 3.919 | 45.369 | 58.487 |
| (b) | 31 | 2.002 | 23.94 | 5.224 | 45.369 | 58.487 |
| (c) | 20 | 2.901 | 7.499 | 5.024 | 20.348 | 44.923 |
| (d) | 20 | 8 | 8 | 8 | 24 | 32 |



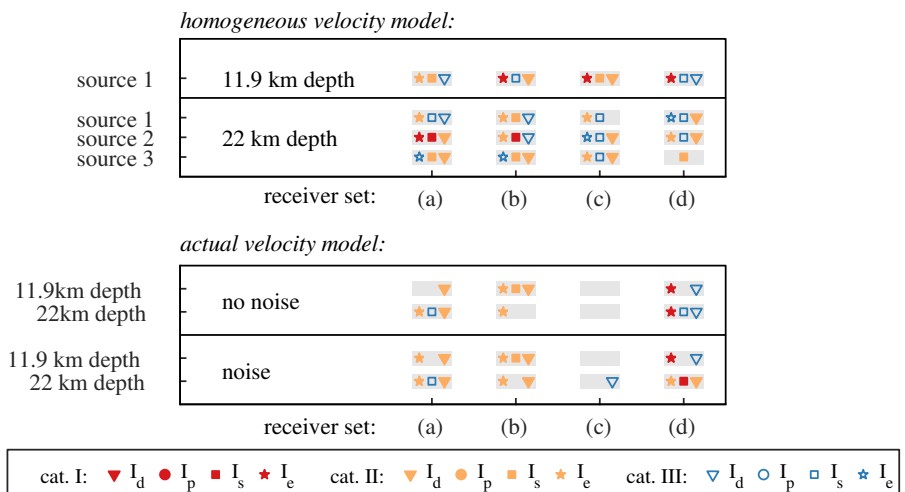

**Figure 15.** The quality of localizations achieved with a strike-slip source in different depths with the different receiver sets and a homogeneous velocity model (top half) and with the velocity model of Zeng et al. (2016) and added noise (SNR = 1)(bottom half). The lower case letters represent the receivers sets of Fig. 14. The localization achieved with the individual imaging conditions were ranked according to the categories from Fig. 2. Grey bars indicate setups that were tested. No symbol in the grey bar means the localization was not successful. Symbol type and colour in the grey bar represent successful localizations with the imaging condition sorted into the respective category.