# Peer review of "Obtaining reliable localizations with Time Reverse Imaging: limits to array design, velocity models and signal-to-noise ratios"

_Solid Earth, 2018_

## Referee Comment (RC1) · Anonymous Referee #1 · 27 Sep 2018

The authors present a good exploration on how to use time inverse imaging to reliably localize seismic events. Enormous synthetic tests on limits to array design, velocity model and signal-to-noise ratios (SNR) have been performed and finally a real dataset in California has been utilized to demonstrate the power of array design based on the synthetic tests. These synthetic tests have shown station distributions and SNR of the data play more important roles than velocity model, even though the true velocity model is really complex like having low velocity zone or fault, which are really impressive to me. But there are some issues I listed below when the array design from the synthetic tests is used to the real data. I think this work is suitable for publication in Solid Earth after the major comments below are addressed.

[Figure]

Major comments are as follows.

1. The study has done a lot of synthetic tests mostly using regular arrays expect for the azimuthal gap and the real dataset. For the azimuthal gap tests, the optimal maximum azimuthal gap is 45o for source at 5 km depth, 47o-52o for source at 11.9 km depth, and 45o-48o for source at 22 km depth (Figure 10). This means more regular arrays should be better than relatively asymmetric arrays. Although the authors proposed an optimal 20-receiver regular array for the real dataset, an irregular 31-receiver array is chosen for the final analysis (Figure 14). Moreover, the optimal 20-receiver regular works better for the actual velocity model than an irregular 20-receiver array, but works a little bit worse for the homogenous velocity model such as source 1 at 11.9 km depth (Figure 15). Since the authors claims that using the previous synthetic tests, one can obtain an optimal array design. The real-data case seems like contrary to the authors' claim. Can the authors present an array design works equally or even better than the 31-receiver irregular array? For example, the authors can add three rows more to the bottom of the 20-receiver regular array, forming a 32-receiver regular array. If this works, the 20-receiver irregular array can be discarded. But if not, the authors need to discuss more about it.

2. The authors improve the RTI procedure by Witten and Artman (2011) using illumination map to remove artifacts from velocity models. To make readers directly know how this works, figures for one velocity model in Figure 11 before dividing illumination map, illumination map, and after dividing could be presented.

3. The authors demonstrate the imaging conditions Ie and Ip in Figure 1. But there other two imaging conditions Is and Id used in this study. To display the imaging conditions well to the readers, the other two should be plotted in Figure 1 as well. At least, I am really interested in the images of the other two imaging conditions.

4. To assess the effect of signal-to-noise ratio to localization quality, a set of SNRs have been used in this study. It will be better to add a noise-free results into Figure 13

as a reference for the other results derived from traces with noise.

In addition, here are some minor comments:

Page 2 Line 8, "Conditions" should be "conditions".

Page 6 Line 25, "position" should be "positions".

Page 14 Line 4, "Fig.(a)" should be "Fig.14(a)".

Page 24 caption of Figure 1. "Conditions" should be "conditions".

Page 25 Figure 2. The width of the source area may need to be labeled such as , where is the wavelength of P wave.

---

## Referee Comment (RC2) · Anonymous Referee #2 · 28 Sep 2018

The paper deals with the capability of the time-reverse imaging method(TRI) for source locations. In particular, it investigates the sensitivity of the method to the network position, size and aperture, velocity model and signal-to-noise ratios. The aim of the work is to define the criteria under which the TRI can be successfully applied. In terms of its aim, the paper would be a significant contribution to this important topic.

The paper is logically organised, but it is not written very clearly - this is the main reason why I suggest a moderate to major revision. My main comments are given below and annotated pdf attached:

1) The paper could be shortened and the main findings presented in a more systematic

[Figure]

way. At times, I have an impression that the authors describe everything what has been done, rather then summarising their main findings. For example, some of the obtained results are very inconclusive (e.g., in some cases the method work better with the less stations or with the noise contaminating signals). My intuitive conclusion would be that it is best to use a random network distribution, spanning a range of inter-station distances with as many stations as possible. I'm not asking explicitly to test such a case, but if it is not too difficult, it could be a good addition

2) "Localization" should be replaced throughout the manuscript with "source location". also, "localization quality" is actually the "location accuracy". It would be good to find a native speaker to read the paper before re-submission if possible.

3) Defining the criteria to assess the performance of TRI is not clear enough. I did not understand what are categories I - IV. Also, some parts are unnecessarily repeated

4) The proposed method is not suitable for shallow sources because the authors mute the upper part of the model. This is an important limitation which should be explicitly stated in the conclusion

5) When the authors are talking about the real data from Southern California, they actually use synthetic data. This is fine but needs to be better explained. Clearly: "To mimic a real case scenario from Southern California, we simulate ............. The advantage of using synthetic data when testing a method is because we know what the true answer is...."

6) The discussion part should be more systematic. It is currently divided by recent literature and it is comparing the results from this study with the literature. Instead, it should be divided by the nature of the results, where the literature is cited as needed.

Taking this points into account (and those from the attached manuscript) would significantly improve the paper and make it a very useful source of the information for the scientific community.

Please also note the supplement to this comment:
https://www.solid-earth-discuss.net/se-2018-76/se-2018-76-RC2-supplement.pdf

**Supplement:**

[revised manuscript text omitted]

---

## Author Comment (AC1) · 3 Nov 2018

Claudia Werner and Erik H. Saenger

claudia.werner@hs-bochum.de

Dear Anonymous Referee 1,

thank you for reviewing our manuscript. We carefully considered all of the comments you made and changed the manuscript accordingly. Thanks to your review we were able to improve the readability of the manuscript significantly. Additionally, we added a new set of simulations based on your suggestion which shows that an array with regular inter-station distances produces more reliable source locations then the random distributed real stations. This and additional changes you suggested are described below in the order of your comments as well as marked in the attached modified manuscript.

[Figure]

Additionally, changes made based on the comments by referee 2 are incorporated as well.

*Please note: In the following text referee's comments are put in bold while the author's response is in normal script. Referee's comments may have been shortened for easier reading. No meaning of content was changed. Numbering of the comments of referees was kept the same.*

**Major comments:**

**1) The study suggests that more regular arrays should be better than relative asymmetric arrays. Can the authors present an array design works equally or even better than the 31-receiver irregular array?**

Thank you for your suggestion. We added extra simulations with 32 regular stations, as you proposed, with the real velocity model in Southern California and found that it produces the most accurate source locations.

**2) The authors improve RTI procedure by Witten and Artman (2011) using illumination map to remove artefacts from velocity models. To make readers directly known how this works, figures for one velocity model in Figure 11 before dividing illumination map, illumination map and after dividing could be presented.**

This is a very good suggestion. However, we feel that the manuscript is quite long already (which was also hinted at by Referee 2) and therefore refrain from adding this additional figure. Nevertheless, we added a more direct reference to the study of Witten and Artman (2011) in which such figures are shown and discussed.

**3) The authors demonstrate the imaging conditions Ie and Ip in Figure 1. But there other two imaging conditions Is and Id used in this study. To display the imaging conditions well to the readers, the other two should be plotted in Figure 1 as well.**

We thank you for pointing out this missing information. We added Id to the figure as

you suggested. As you can see, Id is not able to locate the source reliably in this case. There are artificial focusing spots outside the source region. Since this example shows an explosion source, Is does not show anything and we therefore omitted it.

**4) To assess the effect of signal-to-noise ratio to localization quality, a set of SNRs have been used in this study. It will be better to add a noise-free results into Figure 13 as a reference for the other results derived from traces with noise.**

Thank you for suggesting to add the results without any noise to Figure 13. In our opinion, this will improve the readability.

**Minor comments:**

**Page 2 Line 8: "Conditions" should be "conditions" Page 6 Line 25: "position" should be "positions" Page 14 Line 4: "Fig.(a)" should be "Fig.14(a)"** Thank you for your thorough read of the manuscript. All minor comments as mentioned above were corrected in the revised manuscript as suggested.

**Page 25 Figure 2: The width of the source area may need to be labeled such as where is the wavelength of p-wave**

We regret not designing the figure more self-explanatory. We modified Figure 2 and hope that it is more clear now. We rephrased the x-axis label and added the source position on the x axis. Additionally we marked the y values that are discussed in the text. However, we do not want to put an explicit distance on the x axis because this figure should just show a concept that can be applied to arbitrary models and arbitrary location errors.

Sincerely, Claudia Werner and Erik H. Saenger

Please also note the supplement to this comment:
https://www.solid-earth-discuss.net/se-2018-76/se-2018-76-AC1-supplement.pdf

[Figure]

**Supplement:**

[revised manuscript text omitted]

---

## Author Comment (AC2) · 3 Nov 2018

Claudia Werner and Erik H. Saenger

claudia.werner@hs-bochum.de

Dear Anonymous Referee 2,

thank you for thoroughly studying our manuscript. We would like to especially thank you for the extensive markups done in the supplement to your review. We reviewed all your comments and marked annotations in the pdf and changed the manuscript accordingly. We feel that the overall readability has significantly improved. You suggested to perform an additional simulation with a random network design with random inter-station distances and lots of stations. However, we decided to follow the suggestion of Referee 1 and performed additional simulations with regular inter-station distances

which produce very accurate source locations. This supports our opinion, that an array with regular inter-station distances is superior to an array with irregular inter-station distances if the same amount of stations is used. Often only a limited number of stations is available when conducting field surveys and therefore we feel that it is crucial to understand the importance of the sensitivity of the station placement on the location accuracy. However, we agree with you that a lot of stations improves the location results and therefore as many stations as possible should be used. All changes to the manuscript as suggested are described below in the order of your comments as well as marked in the attached modified manuscript. Additionally, changes made based on the comments by referee 1 are incorporated.

*Please note: In the following text referee's comments are put in bold while the author's response is in normal script. Referee's comments may have been shortened for easier reading. No meaning of content was changed. Numbering of the comments of referees was kept the same.*

**Major comments:**

**1) The paper could be shortened and the main findings presented in a more systematic way. At times, I have an impression that the authors describe everything what has been done, rather then summarizing their main findings.**

Thank you for your thorough review and honest assessment. We rewrote parts of the manuscript to present our main findings more systematically and more clearly. The changes are mentioned below your corresponding comments. However, we were not able to shorten the manuscript significantly without removing a significant part of the content.

**My intuitive conclusion would be that it is best to use a random network distribution, spanning a range of inter-station distances with as many stations as possible. I'm not asking explicitly to test such a case, but if it is not too difficult, it could be a good addition.**

We thank you for sharing your ideas on what would be an optimal station distribution. However, we decided to perform additional simulations with 32 stations with homogeneous inter-station distances and obtained very accurate source locations that are superior to the source locations obtained with the real stations in Southern California. We therefore conclude that with a similar amount of stations, regular inter-station distances are better than random inter-station distances. We believe that this is an important conclusion for the application of TRI to field data. Performing a synthetic simulation as you suggested with a random network distribution with a range of inter-station distances would be an interesting addition to this study. Additionally, we agree with you that a lot of stations improves the location results and therefore as many stations as possible should be used. However, we decided to not include it in order to keep the topic of the manuscript a bit more focused on the applicability. Often there are not a lot of stations available and therefore it is crucial to know that placing stations more regularly will increase location accuracy.

**2) "Localization" should be replaced throughout the manuscript with "source location", also "localization quality" is actually the "location accuracy". It would be good to find a native speaker to read the paper before re-submission if possible.**

As suggested, we discussed the manuscript with expert speakers of the English language and changed "to localize" into "to locate" throughout the manuscript. Additionally "localization quality" was changed to "location accuracy". The word "localization" was either turned into "source location" or kept as "localization" to highlight that we are describing the process of finding a source location in individual phrases. Although not explicitly marked in the supplement, we changed the abstract and the title as well. We thank you for suggesting the term "source location" because now the title incorporates the word "source" and therefore states more directly what the manuscript is dealing with.

**3) Defining the criteria to assess the performance of TRI is not clear enough. I**

**did not understand what are categories I-IV. Also, some parts are unnecessarily repeated.**

We thank you for pointing out that this part is not well explained. We rewrote section "2.3: Evaluation of location success" and concentrated especially on the part where the four categories are explained. Unnecessary repetitions were removed and the explanations were written more precise. Furthermore we updated Figure 2 to help explain this part more directly. Additionally, we highlighted that muting the upper part of the model interferes with locating shallow sources and explained more clearly why we chose a rather large error for the source location.

**4) The proposed method is not suitable for shallow sources because the authors mute the upper part of the model. This is an important limitation and should be explicitly stated in the conclusion.**

We added the muting of the upper part of the model explicitly in the conclusion and stated it more clearly in the discussion. We are much obliged for helping us sharpen our main message.

**5) When the authors are talking about the real data from Southern California, they actually use synthetic data. This is fine but needs to be better explained. Clearly: "To mimic a real case scenario from Southern California, we simulate . . . The advantage of using synthetic data when testing a method is because we know what the true answer is . . ."**

This is a good suggestion. The corresponding paragraph was changed according to your suggestion.

**6) The discussion part should be more systematic. It is currently divided by recent literature and it is comparing the results from this study with the literature. Instead, it should be divided by the nature of the results, where the literature is cited as needed.**

[Figure]

Thank you for your detailed evaluation of our writing style. We did not realize that we accidentally wrote the discussion this way. We rewrote the discussion to make it more clear and removed unnecessary repetitions. We restructured the section to use references from literature to support our findings instead of the other way around. Additionally, we separated the parts of the discussion more clearly into general discussion about the used method, the station distribution with homogeneous velocity models, complex velocity models and noise and finally, future challenges.

**Additional Comments from Referee 2 as found in the supplement to the comment (only major comments are listed, minor comments such as rephrasing of individual sentences or paragraphs for readability are not commented but are marked in the revised manuscript):**

**Suggestion to change some of the section titles** Thank you for suggesting to change some of the section titles. We feel that they more clearly express what the sections are about now. We changed the titles of the following sections according to your suggestions as follows:

section 1.1: We kept the original title (but change the word "localization") to reflect that this section deals only with the source location of TRI and does not discuss the potential of TRI to characterize sources section 3.1: changed according to suggestion section 3.2: changed to be more precise section 3.3: change according to suggestion

**Page 4-5: You suggested to use the deterministic signal case of the signal to produce the illumination map** We thank you for suggesting to further improve this method. However, this was tested in previous studies and also as a predecessor to this study. We found that using the not time reversed signal to produce the illumination map achieved the best results in suppressing artefacts. In our opinion, including an additional part in the manuscript discussing a deterministic signal or random noise to produce the illumination map would be too elaborate and elongate the manuscript unnecessary. However, we added a more direct reference to the paper of Witten and

Artman (2011) where this topic is discussed in more detail.

**Page 5: the mention of source coordinates is meaningless without giving the origin We thank you for pointing out this obvious mistake.**

We changed how we describe the model dimensions at the beginning of the section to include the origin point and to be able to place the source coordinates in relation to the model.

**Page 7, Line 9-10: The referee suggested to use Pythagoras theme to calculate the location error**

We agree with you that Pythagoras' theme should be used for calculation location errors. However, in this study the focus was to find station distributions that yield locations roughly in the vicinity of the initial source location. We therefore did not want to restrict the source area too much. When locating real data it is sometimes better to have a location with a rather large error than no source location at all and we wanted to include this in this study to make the study useful for a wider range of applications. For specific applications, the error should be estimated more accurately of course and the setup should be chosen to minimize the error.

Sincerely, Claudia Werner and Erik H. Saenger

Please also note the supplement to this comment:
https://www.solid-earth-discuss.net/se-2018-76/se-2018-76-AC2-supplement.pdf

**Supplement:**

[revised manuscript text omitted]

---

## Referee Report (RR1)

Comments to Author(s):

The revised version of the paper is significantly improved and the authors responded my comments well. So, I think the revised manuscript is suitable for publication in *Solid Earth*.